# Effect of Acupuncture on Timeliness of Male Shoulder Joint Endurance

**DOI:** 10.3390/ijerph18115638

**Published:** 2021-05-25

**Authors:** I-Lin Wang, Rui Hu, Yi-Ming Chen, Che-Hsiu Chen, Jun Wang, Chun-Sheng Ho

**Affiliations:** 1College of Physical Education, Hubei Normal University, Huangshi 435002, China; ilin@gms.ndhu.edu.tw; 2Graduate Institute, Jilin Sports University, Changchun 130022, China; ruihu0614@gmail.com (R.H.); wjun5980@gmail.com (J.W.); 3Department of Sport Performance, National Taiwan University of Sport, Taichung City 41354, Taiwan; jakic1114@ntupes.edu.tw; 4Division of Physical Medicine and Rehabilitation, Lo-Hsu Medical Foundation, Inc., Lotung Poh-Ai Hospital, Yilan City 26546, Taiwan; 5Department of Physical Therapy, College of Medical and Health Science, Asia University, Taichung City 41354, Taiwan

**Keywords:** acupoints, post-activation potentiation, De Qi, torque, strength duration

## Abstract

Acupuncture as a traditional and commonly used treatment has been used to improve the performance of athletes. In the improvement of female shoulder joint explosive force and muscle endurance also has an immediate effect. However, whether the effect of acupuncture therapy can be maintained after improving athletic performance still worth further discussion. The purpose of this study was to explore the timeless of the physical neurophysiological response induced by acupuncture at specific acupoints in improving endurance performance. Seventeen healthy male participants completed six groups of shoulder joint isokinetic exercises. The isokinetic exercise completed in the first group was taken as the baseline. After acupuncture for 15 min, the following 5 isokinetic experiments were completed. Acupuncture acupoints included Binao (LI14), Jianliao (SJ14), Naohui (SJ13), Zhongfu (LU1), Xiabai (LU4), Tianfu (LU3) and Xiaoluo (SJ12). The results show that acupuncture can improve physical performance for 10–20 min. After acupuncture, the maximum torque, average power, average work and total work values significantly increased (*p* < 0.05). Stimulation of acupoints can effectively improve the performance of periarticular muscle endurance around the shoulder, but this improvement is limited by time.

## 1. Introduction

Acupuncture, the most familiar complementary and alternative therapy to Western medicine, has a history of more than 3000 years in China and is still used to treat diseases and relieve pain [1]. Past studies in the United States and Germany have found that 8–9% of adults use acupuncture to prevent or treat various health problems [2,3]. Therefore, acupuncture is particularly important in treating diseases, relieving pain and restoring health. Past studies found that acupuncture at specific acupoints can induce parasympathetic excitation of the body and produce “De Qi” [4,5,6,7], “De Qi” is thought to be caused by myosin regulating light chain phosphorylation in connective tissue fibroblasts that alters the increased sensitivity of actin and myosin Ca2+ to enhance strengthening of the muscles [8]. Therefore, acupuncture increases the body’s ability to quickly recover from fatigue and improves muscle strength. Furthermore, stimulating the body’s proprioceptive system stimulates neurons to improve motor performance [9]. Acupuncture stimulation of the muscle surface can induce changes in potential in the body to the central nervous system to accelerate limb movements [10] and stimulating neurons in the deeper or shallower layers of the skin causes muscles to contract in a way that strengthens the body and helps improve motor performance [6,7]. Therefore, the use of acupuncture to stimulate the body may cause a rapid response of nerves to be transmitted to the brain and increased muscle strength and motor performance may improve physical muscle endurance.

Athletes stimulate the skin surface to cause muscle contractions that lead to increased strength, resulting in post-activation enhancement (PAP) [11] and the phenomenon of increased muscle strength continues after the cessation of exercise [12]. Acupuncture stimulates the skin of the body and causes nerve excitation, which can accelerate the movement reflex of the limbs and enhance muscle strength [10]. After acupuncture is stopped, the peripheral nerves in the related brain regions are still in the excited state and show continuous strength enhancement [13]. The phenomenon of “De Qi” is similar to PAP, in which the body continues to strengthen after the stimulation is stopped. Therefore, the muscle strength enhancement induced by nerve acupuncture is similar to the PAP phenomenon induced by muscle contraction after skin stimulation because both continue to increase strength and motor performance after the stimulation has ceased. Past studies on the shoulder have found that acupuncture on body surface receptors can increase muscle activity, induce the PAP response and improve muscle endurance and shoulder joint muscle group explosive power [6,7]. Needling skin receptors can cause the physiological response to the PAP phenomenon to enhance motor performance.

Acupuncture may increase the local blood supply to the body, increase cerebral blood flow and adenosine triphosphate (ATP) production, affect motor control (α-motor neurons and γ-motor neurons) and induce different types of neural reflexes [14,15,16]. Acupuncture therapy is effective in relieving fatigue, strengthening muscles and quickly restoring shape [17]. Previous studies have found that acupuncture can improve the muscle tension of female football players, regulate their emotions after the game, eliminate fatigue and improve their athletic performance [16,18]. Acupuncture for 10 min can delay extensor fatigue at 50% of the maximum exertion, increase the maximum torque velocity and peak torque value after the end of extensor exercise and improve physical muscle endurance [19]. Therefore, acupuncture can not only improve the disease, but also improve the performance of endurance. Previous studies that drop jump after acupuncture reduces the risk of injury by reducing ground reactions at a jump height of 50 cm [20]. Many studies have explored the immediate effect of acupuncture intervention and it has been proved that acupuncture is one of the methods that can effectively improve the endurance of shoulder joint muscles. However, the research on its timeliness is still limited. Moreover, whether acupuncture is markedly affected by the rise of “De Qi” remains equivocal as current studies do not investigate whether acupuncture is further impaired with a systemic increase of endurance from baseline.

Therefore, the purpose of this study is to evaluate the cumulative time range of acupuncture to improve shoulder muscle strength, the effects of torque value, power and work value on shoulder muscle endurance of healthy male athletes in different time periods (3 min, 13 min, 23 min, 33 min and 43 min) before and after acupuncture were compared. In conducting this study, we hypothesized that acupuncture could improve shoulder muscular endurance and that this improvement was time-limited, with the performance of acupuncture induced endurance enhancement gradually decreasing over time.

## 2. Materials and Methods

### 2.1. Study Design

Data collection was conducted in the Sports Biomechanics Laboratory of Jilin Sport University. Participants were recruited through online advertising and communication with students after the offline course. All enrolled participants were informed of the purpose and procedure of the study and each subject was required to read and sign the consent form before participating in the study. The present study was approved by the Institutional Ethical Committee of Jilin Sport University (JLSU; Changchun, China; JLSU-IRB no. 2018005). We registered the data in the Chinese Clinical Trial Registry (registration number: ChiCTR1900025414).

### 2.2. Participants

Twenty healthy male Division III athletes aged 18–23 with no shoulder or arm injuries from Jilin Sport University to participate in this study voluntarily. After a preliminary medical examination, three athletes were excluded from the study. Therefore, the actual number of participants observed was 17 (nonathletes; age: 20.1 ± 2.6 years; body mass: 74.5 ± 14.8 kg; height: 177.5 ± 5.0 cm; body mass index (BMI): 22.3 ± 3.1 kg/m^2^; length of their upper arms: 33.6 ± 1.69 cm) participated and completed this study, any neuromusculoskeletal disease, a recent history of acupuncture treatment or use of drugs; drink caffeine or alcohol within 12 h; have not done strenuous physical exercise in 48 years [21]. In addition, participants had no history of musculoskeletal injury to the lateral shoulder joint and their range of motion was within the normal range (adduction–abduction: 0–180°, flexion: 0–170°, extension: 0–40°).

### 2.3. Experimental Protocol

#### 2.3.1. Preparation before the Experiment

After a 5 min warm up on a treadmill at a speed of 8 km/h, the participants laid on an isokinetic dynamometer, all limbs except the shoulder and arm are fixed by a special safety belt to prevent the movement of other parts of the shoulder joint from affecting the movement. An isokinetic dynamometer (Con-Trex MJ; CMV AG, Dübendorf, Switzerland) was used to measure the shoulder motion at a rate of 100 Hz. Flexion/extension of the right shoulder (Figure 1) was tested with participants laying supine on the dynamometer chair and shoulder abduction/adduction of the right arm (Figure 2) was tested with participants in the lateral position on the dynamometer chair. The dynamometer’s axis of rotation and shoulder adaptor height were set with respect to the acromion process and arm length, respectively. The forearm was kept parallel to the shoulder adaptor grasping a handle with the wrist in a neutral position. Participants could release the handle during the rest periods. Participants could release the handle during the rest periods. The participants were required to complete shoulder isokinetic muscle endurance tests, including baseline and posttests of the 5 groups: Post 1, Post 2, Post 3, Post 4 and Post 5.

#### 2.3.2. Start Experiment

Before the experiment, participants completed a group of 5 repetitions isokinetic exercises at 30°/s for shoulder flexion and extension to familiarize procedure. In order to avoid the phenomenon of temporary fatigue, after resting for 5 min, the participants completed a group of 5 shoulder flexion and extension exercises at 60°/s as the pre-acupuncture test (pre) in the formal experiment. Afterwards, an experienced acupuncturist with a health professional qualification certificate issued by the Ministry of Human Resources and Social Security, PRC, and the National Health Commission was invited to participate in the experiment and complete the acupuncture as required. Five post-tests were performed at 3 min (Post 1), 13 min (Post 2), 23 min (Post 3), 33 min (Post 4) and 43 min (Post 5) after acupuncture (the experimental method was the same as that of the Pre). In order to avoid the influence of fatigue, the participants were asked to complete 6 groups of adductive and abductive isokinetic motion tests (1 group pre-test and 5 group post-tests) at 60°/s after a 3-day interval. Figure 3 is the experimental flow.

#### 2.3.3. Acupuncture

The selected acupoints include: LI14, SJ14, SJ13, LU1, LU4, LU3 and SJ12 [7]. These points positions are shown in Figure 4. Perpendicular needling was carried out using sterile disposable needles Vertical acupoints were needled with disposable sterile needles (body points: Dongbang ^®^ 0.3 × 30 mm). The participants accepted acupuncture treatment while sitting. The time of acupuncture was 15 min each time and the depth of acupuncture was 10–25 mm. During this period, techniques such as lifting, inserting and turning were used to stimulate the acupoints so as to generate the sense of “De Qi” [22]. De Qi refers to the sensations of the subject receiving acupuncture, comprising soreness, numbness, heaviness and distention at the site of needle insertion [8]. Acupuncture needles can activate pathways of the nervous system including the central nervous system and the peripheral nervous system [6]. Acupuncture on specific points of meridians can affect the near or far points and improve the performance of muscles around the points [23]. Therefore, we chose the acupoints around the shoulder joint as stimulation points to activate the nerves and muscles around the shoulder joint.

### 2.4. Sample Size Estimation

A priori power analysis (G*Power version 3.1.9.4; Heinrich Heine University Düsseldorf, Düsseldorf, Germany) showed that a minimum of 14 participants was required on the basis of conventional α (0.05) and β (0.80) values and an effect size of 1.55 as in our previous study wang et al. using a similar design (i.e., single group repeated measures analysis with completed full shoulder joint Flex/Ext and Abd/Add movements six times and primary outcome variable (i.e., work completed) [5]. After taking 25% of dropout rate into consideration, the number of participants for this study raise to 17.

### 2.5. Data Processing

Kinetic data were collected by an isokinetic Con-Trex dynamometer (Con-Trex MJ; CMV AG, Dübendorf, Switzerland). The parameters of the shoulder joint include the maximum torque, average power, average work and total work. The torque, work and power can be used as parameters for measuring an increase in force [24]. Acupuncture can improve shoulder muscle endurance and explosive force and average maximum torque, average power, average work and total work were collected through an isokinetic test system to explore the difference before and after acupuncture [6,7].

### 2.6. Statistical Analysis

MATLAB (version R2016a; MathWorks, Inc., Natick, MA, USA) was used to analyze the isokinetic data of the shoulder joint before and after acupuncture. The data are presented as the mean and standard deviation (SD). Repeated one-way ANOVA measurement was used for comparisons between groups (*p* < 0.05). After any significant main effects, pairwise comparisons were made using a Bonferroni post hoc adjustment. The effect size (ES) is demonstrated by Cohen’s d coefficient (Cohen’s D = Mean post−Mean pre/S.D.) that considers 0.2 as a small effect; 0.5 as a medium effect; and 0.8 as a large effect [25]. *p* < 0.05 was set as the level of significance.

## 3. Results

### 3.1. Analysis of Kinematics Data during Flexion and Extension of Shoulder Joint

The results showed that there were significant differences in the maximum torque, average power, average work and total work of the shoulder extension/flexion muscle groups before and after acupuncture (*p* < 0.05) (Figure 5). After acupuncture, the average power of extensor (Figure 5D), the average work of extensor (Figure 5F) and the total work of extensor (Figure 5H) were significantly different at Post 1 and Post 2 (ES varying from 0.9 to 1.75, all *p* < 0.05), indicating that acupuncture could improve the muscle strength of shoulder muscle group for 13 min. The maximum torque of extensor (Figure 5B), the average power of flexor (Figure 5C), the average work of flexor (Figure 5E) and the total work of the flexor muscle (Figure 5G) were significantly different at Post 1, Post 2 and Post 3 (ES varying from 0.9 to 2.05, all *p* < 0.05), indicating that the time of acupuncture improving the muscle strength of the muscle group could last for 23 min. The maximum torques of flexor (Figure 5A) at Post 1, Post 2, Post 3 and Post 4 showed significant differences (ES varying from 0.85 to 1.63, *p* < 0.05), indicating that acupuncture could improve the strength of flexor muscles for 33 min.

### 3.2. Analysis of Kinematics Data during Adduction and Abduction of Shoulder Joint

The results showed that there were significant differences in the maximum torque, average power, average work and total work of the shoulder adduction/abduction muscle groups before and after acupuncture (*p* < 0.05) (Figure 6). After acupuncture, the maximum torque of adductor (Figure 6A), the maximum torque of abductor (Figure 6B), the average work of adductor (Figure 6E), the average work of abductor (Figure 6F), the total work of adductor (Figure 6G) and the total work of abductor (Figure 6H) were significantly different at Post 1, Post 2 and Post 3 (ES varying from 1.10 to 2.45, all *p* < 0.05), indicating that the time of acupuncture improving the muscle strength of the muscle group could last for 23 min. The average power of adductor (Figure 6C) and the average power abductor (Figure 6D) at Post 1, Post 2, Post 3 and Post 4 showed significant differences (ES varying from 0.80 to 2.39, *p* < 0.05), indicating that acupuncture could improve the strength of flexor muscles for 33 min.

## 4. Discussion

This study mainly discusses the time of male shoulder joint endurance after acupuncture at points LI14, SJ14, SJ13, LU1, LU4, LU3 and SJ12. The results show that acupuncture at specific points can increase the maximum torque, average power, average work and total work value. Acupuncture can increase the muscle endurance of the male shoulder joint; muscle strength will be enhanced after at least 10 min of treatment.

### 4.1. Analysis of the Maximum Torque of Shoulder Joint

Maximum torque extension and maximum torque adduction/abduction of the shoulder at Post 1, Post 2 and Post 3 were all higher than the baseline values and the muscle strength of the extensor muscle was increased approximately 23 min after acupuncture. Maximum torque flexion of the shoulder at Post 1, Post 2, Post 3 and Post 4 was higher than that at baseline and the flexor muscle strength increased approximately 33 min after acupuncture. The enhancement of the maximum torque is mainly due to the complementary centripetal/centrifugal contraction of the muscles, which results in increased peak torques when the antagonist and active muscles contract synchronically and improve performance [26]. Past studies have shown that 10 min electrical nerve stimulation of the muscles around the knee joint of endurance athletes can increase the peak limb torque value after exercise and improve the anti-fatigue ability of endurance athletes [19]. Therefore, the increase in muscle torque after acupuncture may be caused by the synchronous contraction of active and antagonistic muscles to improve the body’s anti-fatigue ability and enhance muscle strength. In addition, PAP refers to the physical strength of the body activated by the conditioned reflex activity of the muscle in a short-term sustained increase in the performance of exercise [27]. The muscle performance of rugby players and normal male subjects was improved after stimulation and the PAP effect disappeared 12 min after the last contraction exercise and when the PAP effect was maximum [27,28]. Therefore, there may be a PAP phenomenon to improve muscle performance after acupuncture. After acupuncture, synchronous contractions between muscle groups induced the PAP phenomenon, which increased the torque and sustained strength for approximately 23–33 min.

### 4.2. Analysis of the Average Power of Shoulder Joint

Average power extension of the shoulder at Post 1 and Post 2 was higher than the baseline and the muscle strength of the extensor increased approximately 13 min after acupuncture. Average power flexion of the shoulder at Post 1, Post 2 and Post 3 was higher than the baseline and the muscle strength of the flexor muscle increased approximately 23 min after acupuncture. Average power adduction/abduction of the shoulder at Post 1, Post 2, Post 3 and Post 4 was higher than that at baseline and the flexor muscle strength increased approximately 33 min after acupuncture. Acupoints around the limbs stimulate the “De Qi” phenomenon to affect the neuroendocrine-immune network and change the autonomic nervous function in the body, which is conducive to the recovery of homeostatic bodies [29]. Previous studies have found that electrical stimulation of Jianshi-Neiguan or hegu-Lique acupoints can increase the maximum performance of the limbs after exercise and improve human performance [17]. Therefore, the increase in muscle power after acupuncture may be caused by the “De Qi” response of the body, resulting in an increase in the average power value at approximately 13–33 min of continuous power output.

### 4.3. Analysis of the Average Work of Shoulder Joint

The average work extension and total work extension of the shoulder at Post 1 and Post 2 were all higher than the baseline and the muscle strength of the extensor increased approximately 13 min after acupuncture. Average work flexion, average work adduction/abduction, total work flexion and total work adduction/abduction at Post 1, Post 2 and Post 3 were all higher than baseline and the flexor, adductor and abductor muscle strength increased approximately 23 min after acupuncture. Previous studies have shown that acupuncture at the points around the shoulder joint can trigger the PAP phenomenon and increase the power value after exercise [6]; acupuncture at the Xiaohai and Jianwaishu points for 20 min can activate the trapezius muscle and increase the strength performance of bilateral trapezius muscles [30]; acupuncture at the Zusanli, Sanyinjiao, Qihai and Shenmen points for 15 min can strengthen the quadriceps muscles of recreational athletes [21]. Therefore, acupuncture around the shoulder joint of LI14, SJ14 SJ13, LU1, LU4, LU3 and SJ12 holes can cause the PAP phenomenon to be enhanced in healthy male subjects around the shoulder joint muscle strength and acupuncture can improve the shoulder joint muscle strength endurance at approximately 13–23 min of the average work and the total work increases.

### 4.4. Analysis of the Time of Acupuncture

Maximum torque, average power, average work and total work of shoulder joint flexion/extension and adduction/abductor muscle groups increased after different tests (Post 2, Post 3, Post 4), indicating that there was a continuous strength enhancement performance of the shoulder joint in isokinetic motion for approximately 13–33 min after acupuncture. The study found that the parameter values increased after the completion of the isokinetic motor of the shoulder joint at Post 1, indicating that acupuncture had an immediate effect on improving shoulder muscle strength. Past research has found that needling receptors on the skin cause nerve reflexes to set up the body’s communication mechanism [31], which is transmitted to the brain to regulate muscle activity and increase the recruitment of motor units to increase muscle strength [21]. The brain, cerebellum and limbic regions showed the same activity as the increased muscle activity after acupuncture termination [13]. Therefore, the immediate increase in torque, power and work value of shoulder joint after acupuncture may be related to the internal communication mechanism triggered by acupuncture, which activates the continuous excitation of brain nerves and leads to the immediate enhancement of muscle strength after acupuncture. Moreover, acupuncture increases neural activity that remains elevated for a period of time and may not return to the baseline level during the period of stimulation at rest [13]. Acupuncture may have a time-sensitive effect on improving shoulder joint endurance. Nerve activation still exists during muscle rest, indicating that acupuncture may affect the muscle strength around the shoulder joint, causing it to remain strengthened for a certain period of time after the end of exercise. In addition, acupuncture induced changes in the excitability of the cerebral cortex, which may be related to the location of insertion points [10]. Stimulating specific acupoints can trigger different activation patterns at the cortical level [32]. In this study, the maximum torque, average power, average work and total work of the shoulder flexion/extension and adduction/abduction muscle groups after acupuncture showed no difference at Post 5, indicating that the improvement in muscle strength disappeared after 33 min of exercise. The difference at different times may be due to the different sequence of muscle activity induced by different brain activation patterns triggered by the insertion point of the needle and the acupoint. So far, the past after acupuncture can produce the phenomenon of “De Qi” failing to explain effectively, this research through the observation of the change of the endurance get acupuncture needling phenomenon may be associated with PAP phenomenon; however, there are limitations to this study.

There are limitations to this study. First, this study did not directly analyze and compare the actual movements but only used the isokinetic test instrument to collect and process the changes in the parameters of the moving joint. Second, we did not measure the physiological response or record the electrical activity on the muscle surface to study the physiological mechanism of acupuncture in detail. Thirdly, the research on the timeliness of female shoulder joint acupuncture will be further discussed in the future.

## 5. Conclusions

The current study suggests that the use of acupuncture as replacement therapy could generate the benefits of PAP or “De Qi” to improve shoulder joint endurance. Their use was shown to improve the performance of the upper extremities. In this study, even after the termination of acupuncture, such strength enhancement is still affected by the muscular nerve brought by acupuncture, indicating that the effect of acupuncture on improving muscle strength may be timeliness. Acupuncture at LU1 point for 15 min may enhance the muscle strength of the abductor pectoralis major and pectoralis minor. Acupuncture of LI14, SJ13 and SJ14 for 15 min may enhance the adductor deltoid muscle group. Acupuncture of LU3 and LU4 for 15 min may enhance the muscle strength of flexor biceps brachii. Acupuncture of SJ12 for 15 min may enhance the muscle strength of extensor triceps brachii. Therefore, acupuncture of LU1, LI14, SJ13 and SJ14 can enhance the muscle strength of the adductor and abductor muscles, the time to improve the timeliness of adductor and abductor of shoulder joint muscle strength is about 23–33 min. Acupuncture of LU3, LU4 and SJ12 can activate the muscle strength of flexor and extensor groups, the time to improve the timeliness of flexor and extensor of shoulder joint muscle strength is about 13–33 min. After acupuncture, the body surface receptors may excite the central nervous system, so that the brain is in a state of continuous activation and the threshold of muscular nerve excitation is increased, leading to PAP and “De Qi” phenomena in the body. As a result, even if acupuncture stops, there will still be muscle strength enhancement for a period of time. Therefore, the stimulation effect may fade to disappear with the removal of the silver needle. The current study demonstrates a basis for the application of acupuncture in sports training can be suggested to athletes combine acupuncture in endurance training to improve endurance performance in future studies.

## Figures and Tables

**Figure 1 ijerph-18-05638-f001:**
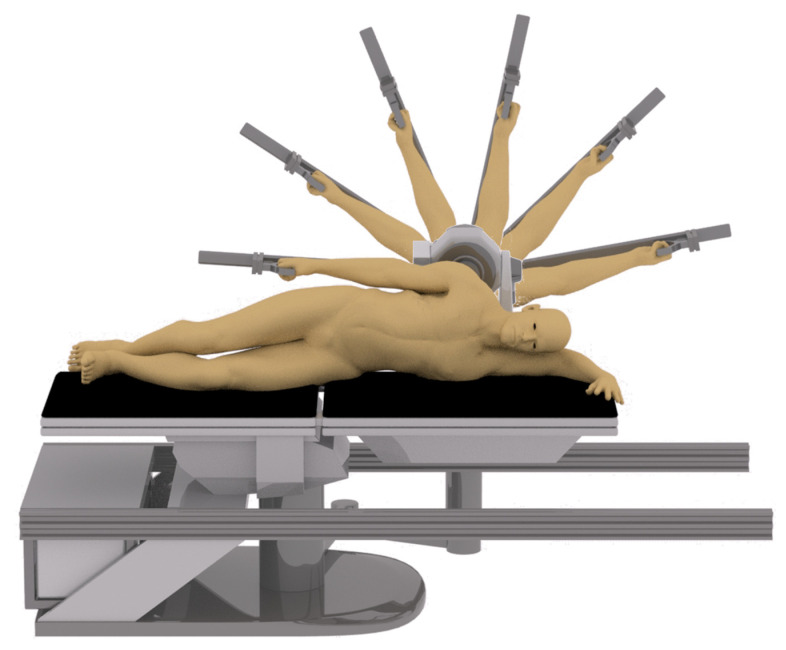
Shoulder flexion/extension isokinetic motion.

**Figure 2 ijerph-18-05638-f002:**
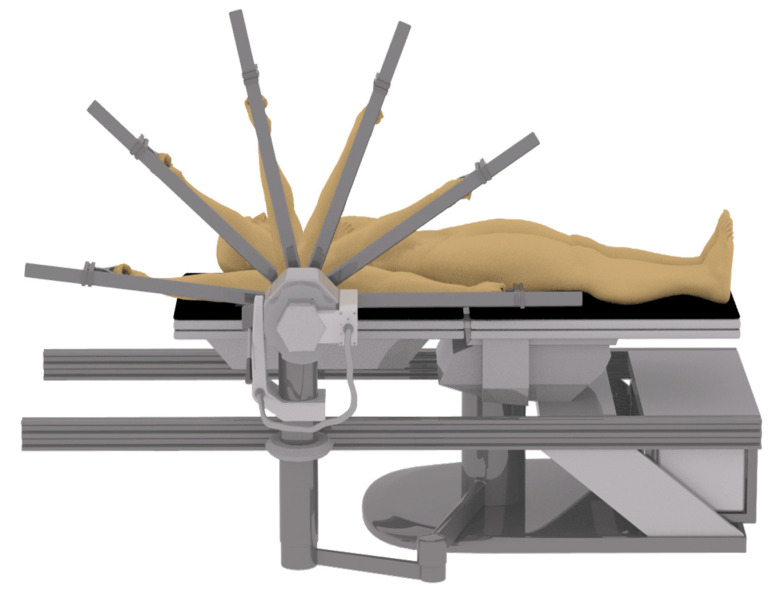
Shoulder adduction/abduction isokinetic motion.

**Figure 3 ijerph-18-05638-f003:**
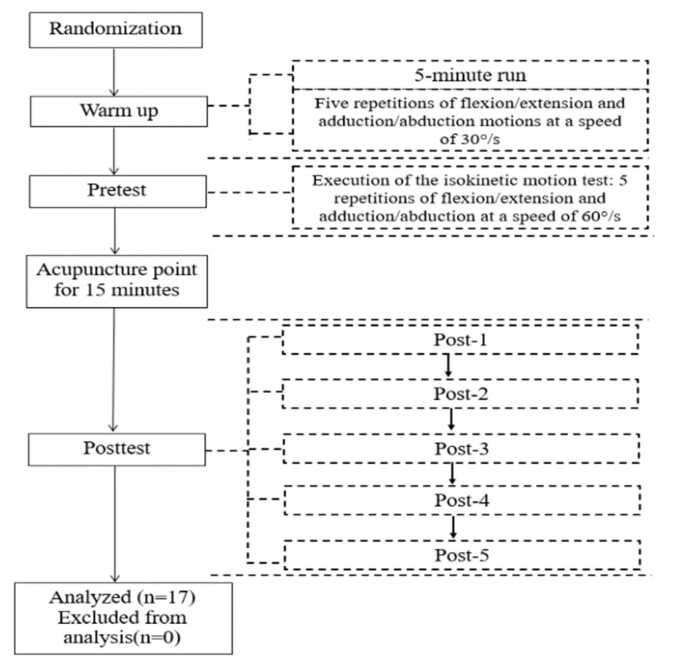
Experimental flow.

**Figure 4 ijerph-18-05638-f004:**
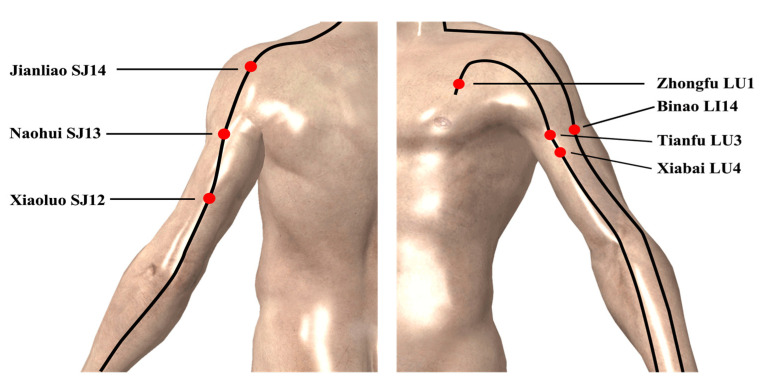
Acupuncture points.

**Figure 5 ijerph-18-05638-f005:**
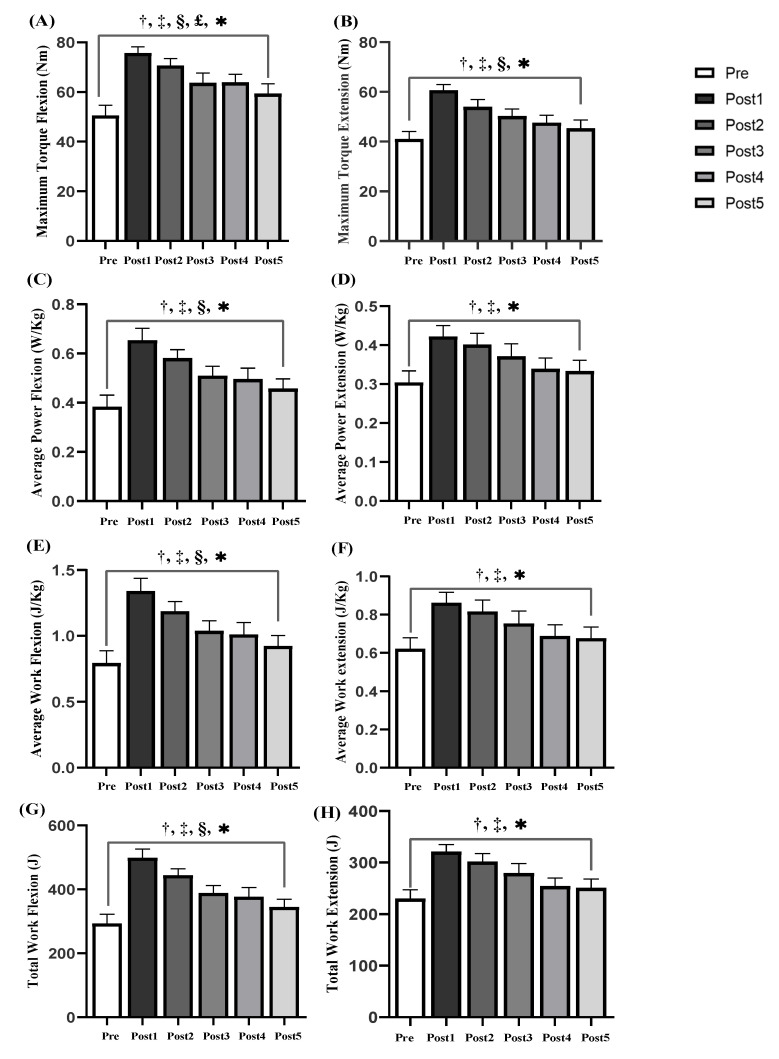
Comparison of flexion and extension (Ext/Flex) motion parameters of shoulder joint before and after acupuncture (*n* = 17): (**A**) Maximum Torque of Flex, (**B**) Maximum Torque of Ext, (**C**) Average Power of Flex, (**D**) Average Power of Ext, (**E**) Average Work of Flex, (**F**) Average Work of Ext, (**G**) Total Work of Flex, (**H**) Total Work of Ext. Note: *: Significant shoulder joint Ext/Flex muscle endurance (*p* < 0.05). †: significant from pre and post1 (*p* < 0.05). ‡: significant from pre and post 2 (*p* < 0.05). §: significant from pre and post 3 (*p* < 0.05). £: significant from pre and post 4 (*p* < 0.05).

**Figure 6 ijerph-18-05638-f006:**
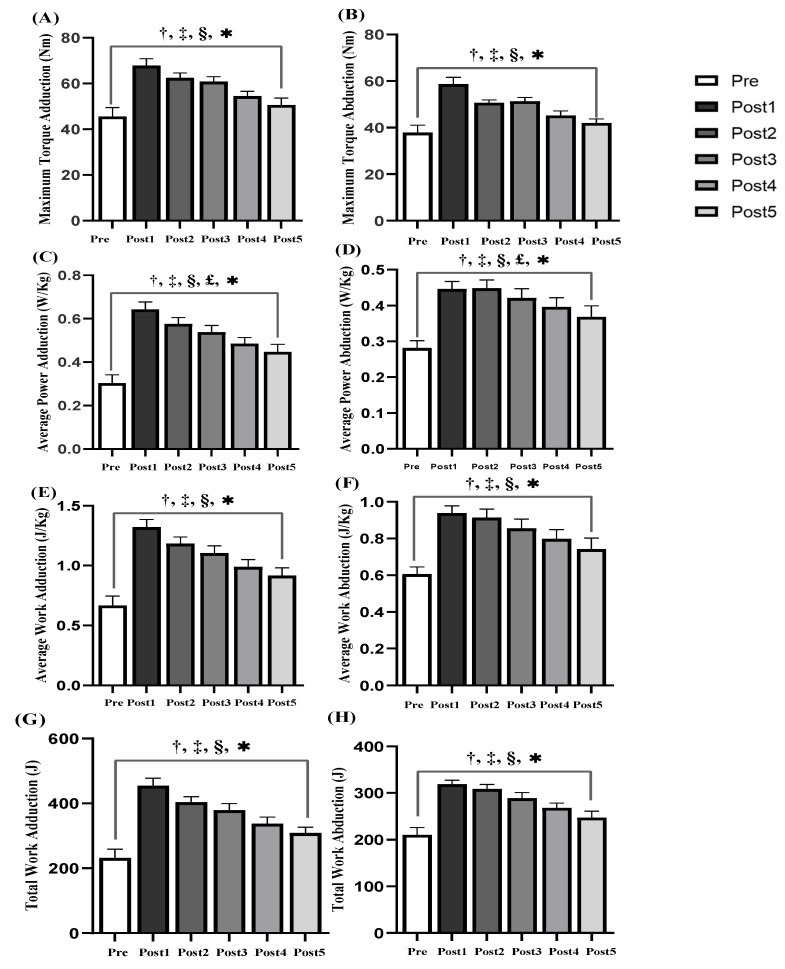
Comparison of adduction/abduction (Add/Abd) motion parameters of shoulder joint before and after acupuncture (*n* = 17): (**A**) Maximum Torque of Add, (**B**) Maximum Torque of Abd, (**C**) Average Power of Add, (**D**) Average Power of Abd, (**E**) Average Work of Add, (**F**) Average Work of Abd, (**G**) Total Work of Add, (**H**) Total Work of Abd. Note: *: Significant shoulder joint Add/Abd muscle endurance (*p* < 0.05). †: significant from pre and post1 (*p* < 0.05). ‡: significant from pre and post 2 (*p* < 0.05). §: significant from pre and post3 (*p* < 0.05). £: significant from pre and post 4 (*p* < 0.05).

## Data Availability

The data used to support the findings of this study are available from the corresponding author upon request.

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
