# Peer review of "Effect of Acupuncture on Timeliness of Male Shoulder Joint Endurance"

_ijerph, 2021, doi:10.3390/ijerph18115638_

Round 1

Reviewer 1 Report

Dear authors, It has been a pleasure to review your paper entitled "Effect of acupunture on teh timeless of endurance of the shoulder joint muscle in males".

I do not have great comments for you, I find the work done of great interest. Just small comments:

Regarding de chosen participants (in Material and Methods section): You write that they are chosen at random, how has that process been? You must explain it. 

The physical activity of the chosen ones has not been taken into account. Will a handball player have the same result as a runner? Why was the female population not included? It could be a limitation of the study. 

Thank you and congratulations for the work done. 

Author Response

Comment: Regarding de chosen participants (in Material and Methods section): You write that they are chosen at random, how has that process been? You must explain it. 

Respond: Thank you for your comment. We have altered the methods paragraph. Please refer to line 151-152.

Original sentence:

The participants were randomly selected, and each participant was required to complete shoulder isokinetic muscle endurance tests, including baseline and posttests of the 5 groups, Post 1, Post 2, Post 3, Post 4, and Post 5.

Altered sentence:

The participants were required to complete shoulder isokinetic muscle endurance tests, including baseline and posttests of the 5 groups, Post 1, Post 2, Post 3, Post 4, and Post 5.

Comment: The physical activity of the chosen ones has not been taken into account. (a). Will a handball player have the same result as a runner? (b). Why was the female population not included? It could be a limitation of the study. 

Respond (a): In this study, only healthy male subjects without shoulder and arm joint injury were studied, and the subject types were not clearly to Whether the handball players have the same effect as the general population after acupuncture needs to be further explored. So this will be taken into account as a limitation of our research in future studies. But a handball player who can make a whiting ball movement may be similar to the action in this study.

Respond (b): The study on the timeliness of acupuncture on female 's shoulder endurance will be observed in the further, and this point will be included in our study limitation. Please refer to line 428-429.

Altered sentence:

There are limitations to this study. First, this study did not directly analyze and compare the actual movements but only used the isokinetic test instrument to collect and process the changes in the parameters of the moving joint. Second, we did not measure the physiological response or record the electrical activity on the muscle surface to study the physiological mechanism of acupuncture in detail. Thirdly, the research on the timeliness of female shoulder joint acupuncture will be further discussed in the future.

Reviewer 2 Report

Estimated Authors,

Estimated Editors,

I've read with great interest the paper from Wang et al. reporting on their studies about the effect of acupuncture on the timeless of endurance of the shoulder joint muscle in males.

Acupuncture belongs to the Traditional Chinese Medicine, representing its key component. Even though it is often defined a sort of "pseudoscience" (please be aware that the present reviewer does not share this point-of-view), available evidence remains, to date, largely inconclusive - therefore, studies like the present one represent a significant contribution to the ongoing debate on the potential acceptance of acupuncture among the "modern medicine".

I would like to stress that the following analysis has nothing to do with acupuncture "per se", and I will focus on the shortcomings of the paper that must be fixed before a further evaluation for a potential acceptance.

More precisely:

1) inclusion criteria are clear, while the reference population is not: Authors write that "Seventeen healthy volunteer males from Jilin Sport University". We know very scant information about their general status (i.e. what about their level of physical activity?). What about their BMI, the length of their upper arms and their basic condition (e.g. the range of movement of the shoulder) Moreover: why only males? And again: why 17 participants? Is it a convenience sample? Please explain such points in the METHODS section and then include, at the beginning of the results section, a detailed description of the general characteristics of the study participants including age, height, weight, measurements of the upper arm, range of motion; because of the reduced number of participants, please include not only average and SD but also median and range.

2) please report (not only in the methods section but also in the caption of the figures) the actual number of observations you made per participant.

3) Results are quite difficult to read as they are crowded with data, that mostly represent a repetition of informations reported in figures. Moreover, Authors have made an extensive use of the symbol "delta" (△) that - because of the repetitive information they have reported, could be avoided improving the readability of the text.

4) Please include how you calculated the effect size. At the knowledge of this reviewer, various formulas actually do exist.

5) Limitation section in an experimental study represents a very important stage of data reporting. In the present form, your paper lacks an appropriate discussion of the limits of the study, and should be expanded in order to accurately address the topic that you (however) have correctly identified. Moreover, you should discuss whether the sample size is sufficient to draw appropriate and general conclusions

6) As the international status of the acupuncture is more discussed than that acknowledged by study Authors, a more critical appraisal in discussion and introduction is, in my opinion, warmly recommended.

7) Eventually, the annex material including the CV of the Authors is not required and could be excluded. Actual photos of the participants / machineries would be more appropriate and interesting to the readers.

Author Response

 Comment: inclusion criteria are clear, while the reference population is not: Authors write that "Seventeen healthy volunteer males from Jilin Sport University". We know very scant information about their general status (i.e. what about their level of physical activity?). What about their BMI, the length of their upper arms and their basic condition (e.g. the range of movement of the shoulder) (a). Moreover: why only males? And again: why 17 participants? Is it a convenience sample? (b). Please explain such points in the METHODS section and then include, at the beginning of the results section, a detailed description of the general characteristics of the study participants including age, height, weight, measurements of the upper arm, range of motion; because of the reduced number of participants, please include not only average and SD but also median and range.

Respond (a): Thank you for your comment. The sentence has been revised. “Twenty healthy male Division III athletes aged between 18-23 with no shoulder or arm injuries from Jilin Sport University to participate in this study voluntarily. After a preliminary medical examination, three athletes were excluded from the study. Therefore, the actual number of participants observed was 17”. Please refer to line 112-115.

Original sentence:

Participants

Seventeen healthy volunteer males from Jilin Sport University (age: 20.1±2.6 years; body mass: 77.7±18.0 kg; and height: 177.5±5.0 cm) were recruited. Exclusion criteria were as follows: cardiovascular, pulmonary, metabolic, neurological, psychiatric and musculoskeletal diseases, hemophilia, muscle soreness, pain, acupuncture treatment within the last 4 weeks and the use of any medications; beverages containing caffeine or alcohol were not allowed for at least 12 h prior to the measurements, and subjects were instructed to maintain their normal level of physical activity during the entire study period and refrain from any form of physical exercise for at least 48 h prior to the tests [1].

Altered sentence:

Participants

Twenty healthy male Division III athletes aged between 18-23 with no shoulder or arm injuries from Jilin Sport University to participate in this study voluntarily. After a preliminary medical examination, three athletes were excluded from the study. Therefore, the actual number of participants observed was 17 (nonathletes; age: 20.1±2.6 years; body mass: 74.5±14.8 kg; height: 177.5±5.0 cm; Body Mass Index (BMI): 22.3±3.1 Kg/m2; length of their upper arms: 33.6±1.69 cm) participated and completed this study. Exclusion criteria were as follows: cardiovascular, pulmonary, metabolic, neurological, psychiatric, and musculoskeletal diseases, hemophilia, muscle soreness, pain, acupuncture treatment within the last 4 weeks and the use of any medications; beverages containing caffeine or alcohol were not allowed for at least 12 h prior to the measurements. Participants had no history of musculoskeletal injury to the lateral shoulder joint, and their range of motion was within the normal range (adduction-abduction: 0°-180°, flexion: 0°-170°, extension: 0°-40°). All participants were instructed to maintain their normal level of physical activity during the entire study period and refrain from any form of physical exercise for at least 48 h prior to the tests [1].

Respond (b): Thank you for your comment. The sentence has been revised.  Please refer to line 115-117 and 125-127. We have added “Sample size estimation” paragraph in the Methods section. Please refer to line 214-220.

Original sentence:

Participants

Seventeen healthy volunteer males from Jilin Sport University (age: 20.1±2.6 years; body mass: 77.7±18.0 kg; and height: 177.5±5.0 cm) were recruited. Exclusion criteria were as follows: cardiovascular, pulmonary, metabolic, neurological, psychiatric and musculoskeletal diseases, hemophilia, muscle soreness, pain, acupuncture treatment within the last 4 weeks and the use of any medications; beverages containing caffeine or alcohol were not allowed for at least 12 h prior to the measurements, and subjects were instructed to maintain their normal level of physical activity during the entire study period and refrain from any form of physical exercise for at least 48 h prior to the tests [1].

Altered sentence:

Participants

Twenty male college students aged between 18-23 with no shoulder or arm injuries from Jilin Sport University to participate in this study voluntarily. After a preliminary medical examination, three athletes were excluded from the study. Therefore, a total of seventeen healthy subjects (nonathletes; age: 20.1±2.6 years; body mass: 74.5±14.8 kg; height: 177.5±5.0 cm; Body Mass Index (BMI): 22.3±3.1 Kg/m2 ; length of their upper arms: 33.6±1.69 cm) participated and completed this study. Exclusion criteria were as follows: cardiovascular, pulmonary, metabolic, neurological, psychiatric, and musculoskeletal diseases, hemophilia, muscle soreness, pain, acupuncture treatment within the last 4 weeks and the use of any medications; beverages containing caffeine or alcohol were not allowed for at least 12 h prior to the measurements. Subjects had no history of musculoskeletal injury to the lateral shoulder joint, and their range of motion was within the normal range (adduction-abduction: 0°-180°, flexion: 0°-170°, extension: 0°-40°). All subjects were instructed to maintain their normal level of physical activity during the entire study period and refrain from any form of physical exercise for at least 48 h prior to the tests [1].

Added sentence:

Sample size estimation

A priori power analysis (G*Power version 3.1.9.4; Heinrich Heine University Düsseldorf, Düsseldorf, Germany) showed that a minimum of 14 participants was required on the basis of conventional α (0.05) and β (0.80) values, and an effect size of 1.55 as in our previous study wang et al using a similar design (i.e., single group repeated measures analysis with completed full shoulder joint Flex/Ext and Abd/Add movements six times and primary outcome variable (i.e., work completed) [2]. After taking 25% of dropout rate into consideration, the number of participants for this study raise to 17.

Comment: please report not only in the methods section but also in the caption of the figures) the actual number of observations you made per participant.

Response: Thank you for your comment. The sentence has been revised.  Please refer to line 115, 281 and 320.

Original sentence:

Participants.

Seventeen healthy volunteer males from Jilin Sport University (age: 20.1±2.6 years; body mass: 77.7±18.0 kg; and height: 177.5±5.0 cm) were recruited. Exclusion criteria were as follows: cardiovascular, pulmonary, metabolic, neurological, psychiatric and musculoskeletal diseases, hemophilia, muscle soreness, pain, acupuncture treatment within the last 4 weeks and the use of any medications; beverages containing caffeine or alcohol were not allowed for at least 12 h prior to the measurements, and subjects were instructed to maintain their normal level of physical activity during the entire study period and refrain from any form of physical exercise for at least 48 h prior to the tests [1].

Figures.

Figure 5: Isokinetic parameters and post hoc test differences in shoulder joint extension/flexion (Ext/Flex) before and after acupuncture. Note: *: Significant shoulder joint Ext/Flex muscle endurance (P<0.05). †: significant from pre and post1 (p < 0.05). ‡: significant from pre and post2 (p < 0.05). §: significant from pre and post3 (p < 0.05). £: significant from pre and post4 (p < 0.05).  

Figure 6: Isokinetic muscle force parameters and post hoc test differences in shoulder joint adduction/abduction (Add/Abd) before and after acupuncture. Note: *: Significant shoulder joint Add/Abd muscle endurance (P<0.05). †: significant from pre and post1 (p < 0.05). ‡: significant from pre and post2 (p < 0.05). §: significant from pre and post3 (p < 0.05). £: significant from pre and post4 (p < 0.05).

Altered sentence:

Participants.

Twenty male college students aged between 18-23 with no shoulder or arm injuries from Jilin Sport University to participate in this study voluntarily. After a preliminary medical examination, three athletes were excluded from the study. Therefore, the actual number of subjects observed was 17 (nonathletes; age: 20.1±2.6 years; body mass: 74.5±14.8 kg; height: 177.5±5.0 cm; Body Mass Index (BMI): 22.3±3.1 Kg/m2 ; length of their upper arms: 33.6±1.69 cm). Exclusion criteria were as follows: cardiovascular, pulmonary, metabolic, neurological, psychiatric, and musculoskeletal diseases, hemophilia, muscle soreness, pain, acupuncture treatment within the last 4 weeks and the use of any medications; beverages containing caffeine or alcohol were not allowed for at least 12 h prior to the measurements. Subjects had no history of musculoskeletal injury to the lateral shoulder joint, and their range of motion was within the normal range (adduction-abduction: 0°-180°, flexion: 0°-170°, extension: 0°-40°). All subjects were instructed to maintain their normal level of physical activity during the entire study period and refrain from any form of physical exercise for at least 48 h prior to the tests [1].

Figures. Please refer to line 217 and 241

Figure 5: Isokinetic parameters and post hoc test differences in shoulder joint extension/flexion (Ext/Flex) before and after acupuncture (n=17). Note: *: Significant shoulder joint Ext/Flex muscle endurance (P<0.05). †: significant from pre and post1 (p < 0.05). ‡: significant from pre and post2 (p < 0.05). §: significant from pre and post3 (p < 0.05). £: significant from pre and post4 (p < 0.05).

Figure 6: Isokinetic muscle force parameters and post hoc test differences in shoulder joint adduction/abduction (Add/Abd) before and after acupuncture (n=17). Note: *: Significant shoulder joint Add/Abd muscle endurance (P<0.05). †: significant from pre and post1 (p < 0.05). ‡: significant from pre and post2 (p < 0.05). §: significant from pre and post3 (p < 0.05). £: significant from pre and post4 (p < 0.05).

Comment: Results are quite difficult to read as they are crowded with data, that mostly represent a repetition of information reported in figures. Moreover, Authors have made an extensive use of the symbol "delta" (△) that - because of the repetitive information they have reported, could be avoided improving the readability of the text.

Response: Thank you for your comment. According to the reviewer suggestion, we have deleted no significant "delta" (). Please refer to line 264-227 and 306-316.

Original sentence:

3.1. Kinematic index of shoulder joint flexion and extension.

Changes in the isokinetic index were found during flexion and extension of the shoulder joint (Figure 5). The maximum torque, average power, average work, and total work increased significantly overall (all p < 0.05) across the flexion and extension values of the shoulder joint after acupuncture, and post hoc analysis revealed significant differences between pre- and post-acupuncture. Our results show that the average power extension was +△38.82% and +△31.91% (all p<0.05,ES=1.75 and 1.02) after acupuncture (Figure 5D); average work extension was +△38.81% and +△31.40% (all p<0.05, ES=1.52 and 0.90) after acupuncture (Figure 5F); total work extension was +△39.53% and +△31.26% (all p<0.05, ES=1.57 and 0.97) after acupuncture (Figure 5H) during prepost1 and post2. The value of the maximum torque extension was +△47.66%,+△31.53% and +△22.36% (all p<0.05, ES varying from 0.90 to1.70) after acupuncture (Figure 5B); average power flexion was +△70.31%, +△51.30% and +△32.81% (all p<0.05,ES varying from 1.05 to 2.05) after acupuncture (Figure 5C); average work flexion was +△69.07%,+△49.75% and +△30.93% (all p<0.05,ES varying from 0.94 to1.81) after acupuncture (Figure 5E); total work flexion was +△70.45%,+△51.54% and +△32.67% (all p<0.05,ES varying from 1 to1.92) after acupuncture (Figure 5G) during pre, post1, post2 and post3. The maximum torque flexion values were +△49.70%, +△39.92%, +△25.97% and +△26.35% (all p<0.05, ES varying from 0.85 to 1.63) after acupuncture (Figure 5A) during prepost1, post2, post3 and post4.

3.2 Kinematic index of shoulder joint of adduction and abduction

Changes in the isokinetic index were found during adduction and abduction of the shoulder joint (Figure 6). The maximum torque, average power, average work, and total work increased significantly overall (all p < 0.05) across the adduction and abduction values of the shoulder joint after acupuncture, and post hoc analysis revealed significant differences pre- and post-acupuncture. Our results show that maximum torque adduction was +△48.22%,+△36.54% and +△32.96% (all p<0.05,ES varying from 1.19 to 2.12) after acupuncture (Figure 6A); maximum torque abduction was +△54.98%, +△33.72% and +△35.52% (all p<0.05,ES varying from 1.30 to 2.26) after acupuncture (Figure 6B); average work adduction was +△98.20%,+△77.36% and +△65.82% (all p<0.05,ES varying from 1.31 to 2.32) after acupuncture (Figure 6E); average work abduction was +△54.95%,+△50.83% and +△41.09% (all p<0.05,ES varying from 1.10 to 1.71) after acupuncture (Figure 6F); total work adduction was +△95.65%,+△73.76% and +△63.23% (all p<0.05,ES varying from 1.37 to 2.45) after acupuncture (Figure 5G); total work abduction was +△51.60%,+△46.73% and +△37.37% (all p<0.05,ES varying from 1.13 to 1.96) after acupuncture (Figure 6H) during prepost1, post2 and post3. The value of the average power adduction was +△112.21%,+△90.10%, +△77.56% and +△60.40% (all p<0.05,ES varying from 0.80 to 2.39) after acupuncture (Figure 6C); average power abduction was +△58.16%,+△58.87%, +△49.29% and +△40.43% (all p<0.05, ES varying from 0.90 to1.68) after acupuncture (Figure 6D) during prepost1, post2, post3 and post4.

Altered sentence:

3.1. Kinematic index of shoulder joint flexion and extension.

The results showed that there were significant differences in the maximum torque, average power, average work and total work of the shoulder extension/flexion muscle groups before and after acupuncture (P < 0.05) (Figure 5). After acupuncture, the average power of extensor (Figure 5D), the average work of extensor (Figure 5F) and the total work of extensor (Figure 5H) were significantly different at Post1 and Post2 (ES varying from 0.9 to 1.75,all P < 0.05), indicating that acupuncture could improve the muscle strength of shoulder muscle group for 13min.The maximum torque of extensor (Figure 5B), the average power of flexor (Figure 5C), the average work of flexor (Figure 5E) and the total work of the flexor muscle (Figure 5G) were significantly different at Post1, Post2 and Post3 (ES varying from 0.9 to 2.05, all P < 0.05), indicating that the time of acupuncture improving the muscle strength of the muscle group could last for 23min.The maximum torques of flexor (Figure 5A) at Post1, Post2, Post3 and Post4 showed significant differences (ES varying from 0.85 to 1.63, P < 0.05), indicating that acupuncture could improve the strength of flexor muscles for 33min.

3.2 Kinematic index of shoulder joint of adduction and abduction

The results showed that there were significant differences in the maximum torque, average power, average work and total work of the shoulder adduction/abduction muscle groups before and after acupuncture (P < 0.05) (Figure 6). After acupuncture, the maximum torque of adductor (Figure 6A), the maximum torque of abductor (Figure 6B), the average work of adductor (Figure 6B), the average work of adductor (Figure 6E), the average work of abductor (Figure 6F), the total work of adductor (Figure 5G), and the total work of abductor (Figure 6H) were significantly different at Post1, Post2 and Post3 (ES varying from 1.10 to 2.45, all P < 0.05), indicating that the time of acupuncture improving the muscle strength of the muscle group could last for 23min. The average power of adductor (Figure 6C) and the average power abductor (Figure 6D) at Post1, Post2, Post3 and Post4 showed significant differences (ES varying from 0.80 to 2.39, P < 0.05), indicating that acupuncture could improve the strength of flexor muscles for 33min.

Comment: Please include how you calculated the effect size. At the knowledge of this reviewer, various formulas actually do exist.

Response: Thank you for your comment. We have added the calculation formula of effect size in the Method section. Please refer to line 241-243.

Original sentence:

Statistical analysis

MATLAB (version R2016a; MathWorks, Inc., Natick, MA) was used to analyze the isokinetic data of the shoulder joint in flexion/extension and adduction/abduction before and after acupuncture. The data are presented as the mean and standard deviation (SD). Repeated one-way ANOVA measurement was used for comparisons between groups (p<0.05). After any significant main effects, pairwise comparisons were made using a Bonferroni post hoc adjustment. Additionally, the effect size (ES) was calculated using MATLAB software (version R2016a; MathWorks, Inc., Natick, MA) and interpreted based on the following criteria: ˂ 0.20 = trivial; 0.20-0.59 = small; 0.60 -1.19 = moderate; 1.20-2.0 = large; ˃ 2.0 = very large. p<0.05 was set as the level of significance.

Altered sentence:

Statistical analysis

MATLAB (version R2016a; MathWorks, Inc., Natick, MA) was used to analyze the isokinetic data of the shoulder joint in flexion/extension and adduction/abduction before and after acupuncture. The data are presented as the mean and standard deviation (SD). One-way Repeated Measures ANOVA was used for comparisons between groups (P<0.05). After any significant main effects, pairwise comparisons were made using a Bonferroni post hoc adjustment. The effect size is demonstrated by Cohen's d coefficient (Cohen’s D=Mean post-Mean pre/S.D.) that considers 0.2 as a small effect; 0.5 as a medium effect; and 0.8 as a large effect [3]. P<0.05 was set as the level of significance.

Comment: Limitation section in an experimental study represents a very important stage of data reporting. (a). In the present form, your paper lacks an appropriate discussion of the limits of the study, and should be expanded in order to accurately address the topic that you (however) have correctly identified. (b). Moreover, you should discuss whether the sample size is sufficient to draw appropriate and general conclusions.

Response (a): Thank you for your comment. We have extended the research restrictions in the experimental study and reconsidered the research scope, and the modifications are as follows: (Please refer to line 428-429).

Original sentence:

There are limitations to this study. First, this study did not directly analyze and compare the actual movements but only used the isokinetic test instrument to collect and process the changes in the parameters of the moving joint. Second, we did not measure the physiological response or record the electrical activity on the muscle surface to study the physiological mechanism of acupuncture in detail.

Altered sentence:

There are limitations to this study. First, this study did not directly analyze and compare the actual movements but only used the isokinetic test instrument to collect and process the changes in the parameters of the moving joint. Second, we did not measure the physiological response or record the electrical activity on the muscle surface to study the physiological mechanism of acupuncture in detail. Thirdly, the research on the timeliness of female shoulder joint acupuncture will be further discussed in the future.

Response (b): Thank you for your comment. The sentence has been added. Please refer to line 213-220.

Add sentence:

A priori power analysis (G*Power version 3.1.9.4; Heinrich Heine University Düsseldorf, Düsseldorf, Germany) showed that a minimum of 14 participants was required on the basis of conventional α (0.05) and β (0.80) values, and an effect size of 1.55 as in our previous study wang et al using a similar design (i.e., single group repeated measures analysis with completed full shoulder joint Flex/Ext and Abd/Add movements six times and primary outcome variable (i.e., work completed) [2]. After taking 25% of dropout rate into consideration, the number of participants for this study raise to 17.

Comment: As the international status of the acupuncture is more discussed than that acknowledged by study Authors, a more critical appraisal in discussion and introduction is, in my opinion, warmly recommended.

Response: Thank you for your comment. The introduction and discussion paragraph have been altered to critical appraisal. Please refer to line 91-94 and 420-423.

Original sentence:

Introduction- However, there is still a lack of research on the efficacy of acupuncture for moxibustion. To explore the effect of acupuncture on improving sports performance, this study further observed the efficacy of acupuncture in improving endurance performance to apply it to the training of athletes in the future. Therefore, the purpose of this study was to explore the effect of acupuncture on improving its efficacy for shoulder endurance by evaluating shoulder motor performance after acupuncture treatment at different time periods. The study hypothesized that the increase in endurance caused by acupuncture at specific points might weaken after a period of time, and there was a time limit for acupuncture’s ability to prolong shoulder motor performance.

Altered sentence:

Introduction- However, there is still a lack of research on the efficacy of acupuncture for moxibustion. To explore the effect of acupuncture on improving sports performance, this study further observed the efficacy of acupuncture in improving endurance performance to apply it to the training of athletes in the future. Moreover, whether acupuncture is markedly affected by the rise of “De qi” remains equivocal as current studies do not investigate whether acupuncture is further impaired with a systemic increase of endurance from baseline. Therefore, the purpose of this study was to explore the effect of acupuncture on improving its efficacy for shoulder endurance by evaluating shoulder motor performance after acupuncture treatment at different time periods. The study hypothesized that the increase in endurance caused by acupuncture at specific points might weaken after a period of time, and there was a time limit for acupuncture’s ability to prolong shoulder motor performance.

Original sentence:

Disscusion- In this study, the maximum torque, average power, average work and total work of the shoulder flexion/extension and adduction/abduction muscle groups after acupuncture showed no difference at post5, indicating that the improvement in muscle strength disappeared after 20 minutes of exercise. The difference at different times may be due to the different sequence of muscle activity induced by different brain activation patterns triggered by the insertion point of the needle and the acupoint.

Altered sentence:

Dusscusion- In this study, the maximum torque, average power, average work and total work of the shoulder flexion/extension and adduction/abduction muscle groups after acupuncture showed no difference at post5, indicating that the improvement in muscle strength disappeared after 20 minutes of exercise. The difference at different times may be due to the different sequence of muscle activity induced by different brain activation patterns triggered by the insertion point of the needle and the acupoint. So far, the past after acupuncture can produce the phenomenon of “De qi” failing to explain effectively, this research through the observation of the change of the endurance get acupuncture needling phenomenon may be associated with PAP phenomenon, however, there are limitations to this study.

Comment: Eventually, the annex material including the CV of the Authors is not required and could be excluded. Actual photos of the participants / machineries would be more appropriate and interesting to the readers.

Response: Thank you for your comment. We have removed the CV. In our article, there are simulated pictures of subjects doing shoulder joint adduction-abduction and flexion- extension isokinetic exercises

Submission Date

02 April 2021

Date of this review

23 Apr 2021 21:52:04

Reference

  1. Wang, I.L., et al., Effect of Acupuncture on Muscle Endurance in the Female Shoulder Joint: A Pilot Study. Evid. Based Complementary Altern. Med., 2020. 2020: p. 9786367.
  2. Wang, I.L., et al., Effect of Acupuncture on the Timeliness of Explosive Forces Generated by the Male Shoulder Joint. Evid. Based Complementary Altern. Med., 2021. 2021: p. 5585605.
  3. Marchesini Stival, R.S., et al., Acupuncture in fibromyalgia: a randomized, controlled study addressing the immediate pain response. Rev Bras Reumatol, 2014. 54(6): p. 431-436.

Reviewer 3 Report

Dear authors:

The work done is interesting. After the revision I have recommendations and suggestions that will help to improve the manuscript. Please see the attached document.

Author Response

RECOMMENDATIONS FOR AUTHORS

Line

Comment

ACTION

New line

19-32

Please provide a structural summary in a single paragraph. Do not separate the summary into sections.

RESPONSE: Thank you for your comment.

Acupuncture as a traditional and commonly used treatment has been used to improve the performance of athletes. In the improvement of female shoulder joint explosive force and muscle endurance also has an immediate effect. However, whether the effect of acupuncture therapy can be maintained after improving athletic performance remains unclear. The purpose of this study was to explore the timeliness of the physical neurophysiological response induced by acupuncture at specific acupoints in improving endurance performance. Seventeen healthy male participants completed six groups of shoulder joint isokinetic exercises. The isokinetic exercise completed in the first group was taken as the baseline. After acupuncture for 15 minutes, the following 5 isokinetic experiments were completed. Acupuncture acupoints included Binao (LI14), Jianliao (SJ14), Naohui (SJ13), Zhongfu (LU1), Xiabai (LU4), Tianfu (LU3), and Xiaoluo (SJ12). The results show that acupuncture can improve physical performance for 10- 20 minutes. After acupuncture, the maximum torque, average power, average work, and total work values significantly increased (P<0.05). Therefore, the support degree indicates that stimulation of acupoints around the shoulder joint can effectively improve the performance of periarticular muscle endurance, but this improvement is limited by time.

Line 23-36.

33

Do not repeat the keywords already mentioned in the title of the study. I recommend you to change them with relevant words mentioned in the study.

RESPONSE: Thank you for your comment. The keywords in this study have been revised.

Original sentence: Endurance; Acupuncture; the timeless of shoulder joint; sports performance

Altered sentence: Acupoints; Post-activation potentiation; De qi; Torque;Strength duration

Line 37-38

40-41

It states "Previous studies", and only puts a quotation [4].

RESPONSE: Thank you for your comment. This sentence has been revised.

Original sentence: Past studies found that acupuncture at specific acupoints can induce parasympathetic excitation of the body and produce "De Qi" [1-4]

Line 47-48

68

Never specify the acronym "ATP" beforehand, you must give its meaning, so that every reader can understand it. Check throughout the document that the acronyms are clearly described.

RESPONSE: Thank you for your comment. This sentence has been revised.

Original sentence: Acupuncture may increase the local blood supply to the body, increase cerebral blood flow and ATP production, affect motor control (α-motor neurons and γ-motor neurons), and induce different types of neural reflexes [5-7].

Altered sentence: Acupuncture may increase the local blood supply to the body, increase cerebral blood flow and adenosine triphosphate (ATP) production, affect motor control (α-motor neurons and γ-motor neurons), and induce different types of neural reflexes  [5-7].

Line 74

78-86

Rewrite. State clearly what the objective of the study is. What are you going to do, how did you do it, and what did you do it for?

RESPONSE: Thank you for your comment. This sentence has been rewritten.

Original sentence: However, there is still a lack of research on the efficacy of acupuncture for moxibustion. To explore the effect of acupuncture on improving sports performance, this study further observed the efficacy of acupuncture in improving endurance performance to apply it to the training of athletes in the future. Therefore, the purpose of this study was to explore the effect of acupuncture on improving its efficacy for shoulder endurance by evaluating shoulder motor performance after acupuncture treatment at different time periods. The study hypothesized that the increase in endurance caused by acupuncture at specific points might weaken after a period, and there was a time limit for acupuncture’s ability to prolong shoulder motor performance.

Altered sentence: Many studies have explored the immediate effect of acupuncture intervention, and it has been proved that acupuncture is one of the methods that can effectively improve the endurance of shoulder joint muscles. However, the research on its timeliness is still limited. However, the research on its timeliness is still limited. Moreover, whether acupuncture is markedly affected by the rise of “De qi” remains equivocal as current studies do not investigate whether acupuncture is further impaired with a systemic increase of endurance from baseline. Therefore, this study uses isokinetic testing system, selects healthy male subjects, and designs a single group of pre-and post-test control experiments to explore the timeliness of acupuncture in im-proving shoulder joint muscle endurance. The hypothesis is that the improvement of shoulder joint endurance after acupuncture will be affected by time, and the acupuncture effect will gradually decrease with the extension of time after acupuncture.

Line 91-101

87-178

When the investigators clearly determine the objective, the methodology can be arranged. The methodology is not clear. Explain the experimental process clearly, I recommend doing it in steps.

RESPONSE: Thank you for your advice, which I have explained step by step.

2. Materials and Methods

2.1. Study Design

This study was a laboratory randomized controlled trial. Data collection was conducted in the Sports Biomechanics Laboratory of Jilin Sport University. Participants were recruited through online advertising and communication with students after the offline course. All enrolled participants were informed of the purpose and procedure of the study, and each subject was required to read and sign the consent form before participating in the study. The present study was approved by the Institutional Ethical Committee of Jilin Sport University (JLSU; Changchun, China; JLSU-IRB no. 2018005). We registered the data in the Chinese Clinical Trial Registry (registration number: ChiCTR1900025414).

2.2. Participants

Twenty male college students aged between 18-23 with no shoulder or arm injuries from Jilin Sport University to participate in this study voluntarily. After a preliminary medical examination, three athletes were excluded from the study. Therefore, the actual number of subjects observed was 17 (nonathletes; age: 20.1±2.6 years; body mass: 74.5±14.8 kg; height: 177.5±5.0 cm; Body Mass Index (BMI): 22.3±3.1 Kg/m2; length of their upper arms: 33.6±1.69 cm) participated and completed this study. Exclusion criteria were as follows: cardiovascular, pulmonary, metabolic, neurological, psychiatric, and musculoskeletal diseases, hemophilia, muscle soreness, pain, acupuncture treatment within the last 4 weeks and the use of any medications; beverages containing caffeine or alcohol were not allowed for at least 12 h prior to the measurements. Subjects had no history of musculoskeletal injury to the lateral shoulder joint, and their range of motion was within the normal range (adduction-abduction: 0°-180°, flexion: 0°-170°, extension: 0°-40°). All subjects were instructed to maintain their normal level of physical activity during the entire study period and refrain from any form of physical exercise for at least 48 h prior to the tests [2].

2.3. Experimental protocol

Preparation before the experiment

After a 5-min warm up on a treadmill at a speed of 8 km/h, the participants laid on an isokinetic dynamometer, and their lower body, waist, and thighs were stabilized using specific straps to control extraneous body movements. An isokinetic dynamometer (Con-Trex MJ; CMV AG, Dübendorf, Switzerland) was used to measure the shoulder flexion/extension and abduction/adduction torque at a rate of 100 Hz. Flexion/extension of the right shoulder (Figure 1) was tested with participants laying supine on the dynamometer chair, and shoulder abduction/adduction of the right arm (Figure 2) was tested with participants in the lateral position on the dynamometer chair. The seat belt around the pelvis and arm flexed at 90° with an extended elbow. The dynamometer’s axis of rotation and shoulder adaptor height were set with respect to the acromion process and arm length, respectively. The forearm was kept parallel to the shoulder adaptor grasping a handle with the wrist in a neutral position. Participants could release the handle during the rest periods.

Start experiment

Before the experiment, subjects completed a group of 5 repetitions isokinetic exercises at 30°/s for shoulder flexion and extension to familiarize procedure. In order to avoid the phenomenon of temporary fatigue, after resting for 5min, the subjects completed a group of 5 shoulder flexion and extension exercises at 60°/s as the pre-acupuncture test (Pre) in the formal experiment. Afterwards, an experienced acupuncturist who has a health professional qualification certificate approved by the Ministry of Human Resources and Social Security of the People's Republic of China and the National Health Commission uses disposable sterile needles to stimulate selected acupoints around the shoulder joint. Five post-tests were performed at 3min (Post1), 13min (Post2), 23min (Post3), 33min (Post4) and 43min (Post5) after acupuncture (the experimental method was the same as that of the Pre). In order to avoid the influence of fatigue, the subjects were asked to complete 6 groups of adductive and abductive isokinetic motion tests (1 group pre-test and 5 group post-test) at 60°/s after 3 days interval. The experimental flow is shown in figure 3.

Acupuncture7238624

The following classical acupuncture points were used in the listed order: LI14, SJ14, SJ13, LU1, LU4, LU3, and SJ12 [2]. These points positions are shown in Figure 4. Perpendicular needling was carried out using sterile disposable needles (body points: Dongbang® 0.3×30 mm). The participants were in a sitting position, and the piercing needles were inserted vertically 10–25 mm in depth and manually manipulated by lifting, thrusting, and rotating methods with uniform reinforcing-reducing techniques to produce the sensation known as “De qi” [8]. De qi refers to the sensations of the subject receiving acupuncture, comprising soreness, numbness, heaviness, and distention at the site of needle insertion  [9]. The acupuncture technique involves mild reinforcing and attenuating, retaining the needle in the acupoints for 15 min. All the needles inserted in the acupoints need to be in line horizontally and vertically. Acupuncture needles can modulate the activity of peripheral and central neural pathways [3]. Acupuncture at specific acupoints along the meridians exerts therapeutic effects on nearby and/or distant regions, and acupuncture at specific points in the body can improve athletes' muscle performance [10]. Therefore, we chose the acupoints around the shoulder joint as stimulation points to activate the nerves and muscles around the shoulder joint.

Line 102-214

  1. Hori, E., K. Takamoto, et al., Effects of acupuncture on the brain hemodynamics. Auton Neurosci, 2010, 157, 74-80.
  2. Wang, I.L., Y.-M. Chen, et al., Effect of Acupuncture on Muscle Endurance in the Female Shoulder Joint: A Pilot Study. Evid. Based Complementary Altern. Med., 2020, 2020, 9786367.
  3. Wang, I.L., Y.-M. Chen, et al., Effects of Acupuncture on Explosive Force Production by the Healthy Female Shoulder Joint. Evid. Based Complementary Altern. Med., 2020, 2020, 8835672.
  4. Wang, I.L., J. Wang, et al., Effect of Acupuncture on the Timeliness of Explosive Forces Generated by the Male Shoulder Joint. Evid. Based Complementary Altern. Med., 2021, 2021, 5585605.
  5. Kuo, T., C. Lin, and F. Ho, The soreness and numbness effect of acupuncture on skin blood flow. Am. J. Chinese Med., 2004, 32, 117-29.
  6. Newberg, A., P. Lariccia, et al., Cerebral blood flow effects of pain and acupuncture: a preliminary single-photon emission computed tomography imaging study. J Neuroimaging 2005, 15, 43-9.
  7. Sun, D.-l., Y. Zhang, and D.-l. Chen, Research progress in sports fatigue prevented and treated by acupuncture. J. Acupunct. Tuina Sci., 2009, 7, 123-128.
  8. Li, H., D. Long, et al., A clinical study to assess the influence of acupuncture at "Wang's Jiaji" acupoints on limb spasticity in patients in convalescent stage of ischemic stroke: study protocol for a randomized controlled trial. Trials, 2019, 20, 419.
  9. Kong, J., R. Gollub, et al., Acupuncture de qi, from qualitative history to quantitative measurement. J Altern Complement Med, 2007, 13, 1059-1070.
  10. Costa, L. and J. de Araujo, The immediate effects of local and adjacent acupuncture on the tibialis anterior muscle: a human study. Am. J. Chinese Med., 2008, 3, 17.

Round 2

Reviewer 2 Report

Estimated Authors,

I'm congratulating with you for the extensive efforts you paid in order to cope with my requirements (that, I'm perfectly aware of, were not so easy to reply).

I've no further requirements and I endorse the eventual acceptance of this paper.

Author Response

Thank you for your comment. Those comments are all valuable and very helpful for revising and improving our paper.

Reviewer 3 Report

Dear authors, improvements are evident in your manuscript. But certain aspects still need to be improved:

1. The last paragraph of the introduction is still unclear, clearly define the objective of your research. Sort out your wording.

2. The conclusions should be written clearly. They are too brief. It is possible to add an additional paragraph on the scientific contribution, what could be improved and limitations.

Author Response

  1. Comment: The last paragraph of the introduction is still unclear, clearly define the objective of your research. Sort out your wording.

Response: Thank you for your comment. The sentence has been revised.  Please refer to line 82-88.

Original sentence:

Therefore, this study uses isokinetic testing system, selects healthy male subjects, and designs a single group of pre-and post-test control experiments to explore the timeliness of acupuncture in improving shoulder joint muscle endurance. The hypothesis is that the improvement of shoulder joint endurance after acupuncture will be affected by time, and the acupuncture effect will gradually decrease with the extension of time after acupuncture.

Altered sentence:

Therefore, the purpose of this study is to evaluate the cumulative time range of acupuncture to improve shoulder muscle strength, the effects of torque value, power and work value on shoulder muscle endurance of healthy male athletes in different time periods (3min, 13min, 23min, 33min and 43min) before and after acupuncture were compared. In conducting this study, we hypothesized that acupuncture could improve shoulder muscular endurance and that this improvement was time-limited, with the performance of acupuncture induced endurance enhancement gradually decreasing over time.

  1. Comment: The conclusions should be written clearly. They are too brief. It is possible to add an additional paragraph on the scientific contribution, what could be improved and limitations. Thank you for your comment. The sentence has been revised. Please refer to line 336-356.

Original sentence:

Acupuncture stimulates the nerves in the body and induces the muscles in the stimulated area to produce a movement response to the limbs, and can maintain movement activities under the state of strength enhancement for a period of time, which is helpful to improve the performance of physical endurance. The results of the study compared the post1, post2, post3, post4 and post5 values after acupuncture with those before acupuncture and found that the time of improvement of shoulder joint endurance was approximately 13-33 minutes. The muscle performance of the stimulated part was temporarily enhanced, and the activation gradually weakened over time. Acupuncture can enhance nerve excitability, resulting in the PAP phenomenon in the muscles around the shoulder joint, and the strength enhancement remains approximately 13-33 minutes after the cessation of acupuncture. In summary, it is suggested that this research method should be applied to help improve athletes' performance in actual competition situations.

Altered sentence:

 The current study suggests that the use of acupuncture as replacement therapy could generate the benefits of PAP or "De qi" to improve shoulder joint endurance. Their use was shown to improve the performance of the upper extremities. In this study, even after the termination of acupuncture, such strength enhancement is still affected by the muscular nerve brought by acupuncture, indicating that the effect of acupuncture on improving muscle strength may be timeliness. Acupuncture at LU1 point for 15 minutes may enhance the muscle strength of the abductor pectoralis major and pectoralis minor. Acupuncture of LI14, SJ13, and SJ14 for 15minutes may enhance the adductor deltoid muscle group. Acupuncture of LU3 and LU4 for 15 minutes may enhance the muscle strength of flexor biceps brachii. Acupuncture of SJ12 for 15 minutes may enhance the muscle strength of extensor triceps brachii. Therefore, acupuncture of LU1, LI14, SJ13 and SJ14 can enhance the muscle strength of the adductor and abductor muscles, the time to improve the timeliness of adductor and abductor of shoulder joint muscle strength is about 23-33min. Acupuncture of LU3, LU4 and SJ12 can activate the muscle strength of flexor and extensor groups, the time to improve the timeliness of flexor and extensor of shoulder joint muscle strength is about 13-33min. After acupuncture, the body surface receptors may excite the central nervous system, so that the brain is in a state of continuous activation, and the threshold of muscular nerve excitation is increased, leading to PAP and "De qi" phenomena in the body. As a result, even if acupuncture stops, there will still be muscle strength enhancement for a period of time. Therefore, the stimulation effect may fade to disappear with the removal of the silver needle. The current study demonstrates a basis for the application of acupuncture in sports training can be suggested to athletes combine acupuncture with endurance training to improve athletic performance in future studies.